# Maternal Immune Activation Induces Adolescent Cognitive Deficits Preceded by Developmental Perturbations in Cortical Reelin Signalling

**DOI:** 10.3390/biom13030489

**Published:** 2023-03-07

**Authors:** Rebecca M. Woods, Jarred M. Lorusso, Isabella Harris, Hager M. Kowash, Christopher Murgatroyd, Joanna C. Neill, Jocelyn D. Glazier, Michael Harte, Reinmar Hager

**Affiliations:** 1Division of Evolution, Infection and Genomics, School of Biological Sciences, Manchester Academic Health Science Centre, Faculty of Biology, Medicine & Health, University of Manchester, Manchester M139PL, UK; 2Maternal and Fetal Health Research Centre, School of Medical Sciences, Manchester Academic Health Science Centre, Faculty of Biology, Medicine & Health, University of Manchester, Manchester M139PL, UK; 3Department of Life Sciences, Manchester Metropolitan University, Manchester M156BH, UK; 4Division of Pharmacy & Optometry, School of Health Sciences, Manchester Academic Health Science Centre, Faculty of Biology, Medicine & Health, University of Manchester, Manchester M139PL, UK

**Keywords:** maternal immune activation, schizophrenia, Reelin, epigenetics

## Abstract

Exposure to maternal immune activation (MIA) in utero significantly elevates the risk of developing schizophrenia and other neurodevelopmental disorders. To understand the biological mechanisms underlying the link between MIA and increased risk, preclinical animal models have focussed on specific signalling pathways in the brain that mediate symptoms associated with neurodevelopmental disorders such as cognitive dysfunction. Reelin signalling in multiple brain regions is involved in neuronal migration, synaptic plasticity and long-term potentiation, and has been implicated in cognitive deficits. However, how regulation of Reelin expression is affected by MIA across cortical development and associated cognitive functions remains largely unclear. Using a MIA rat model, here we demonstrate cognitive deficits in adolescent object-location memory in MIA offspring and reductions in *Reln* expression prenatally and in the adult prefrontal cortex. Further, developmental disturbances in gene/protein expression and DNA methylation of downstream signalling components occurred subsequent to MIA-induced Reelin dysregulation and prior to cognitive deficits. We propose that MIA-induced dysregulation of Reelin signalling contributes to the emergence of prefrontal cortex-mediated cognitive deficits through altered NMDA receptor function, resulting in inefficient long-term potentiation. Our data suggest a developmental window during which attenuation of Reelin signalling may provide a possible therapeutic target.

## 1. Introduction

Schizophrenia is a heterogeneous neurodevelopmental disorder affecting approximately 21 million individuals globally [1]. Diagnosis is dependent on three primary symptom domains: positive symptoms, which “add” to typical behaviour and include symptoms such as delusions or hallucinations; negative symptoms, which may include flattened affect or social withdrawal; and cognitive dysfunction, specifically in the domains of attention and working memory [2]. Due to the heterogeneity of symptoms and variability of symptom presentation within domains [3], the root aetiology of schizophrenia is still uncertain. Proposed mechanisms include hypofunction of ionotropic glutamate receptors and subsequent dopaminergic hyperfunction as possible contributors to the aetiopathogenesis of the condition [4]. Current treatments consist primarily of the use of typical or atypical antipsychotics which aim to ameliorate positive symptoms via dopaminergic antagonism, but their effects on cognitive dysfunction are negligible or unclear [4,5,6]. The risk of developing schizophrenia may fundamentally be dependent on both genetic (such as genetic variants in *DISC1* [7]) and multiple environmental risk factors but can vary significantly between affected individuals. Here, exposure to maternal infection in utero is considered a key environmental factor that increases the risk for offspring [8].

The use of preclinical models allows for the control of experimental variables when investigating environmental risk factors such as exposure to prenatal infection and associated maternal immune activation (MIA). These animal models enable neuroanatomical region- and age-specific electrophysiological, genetic, epigenetic or proteomic analyses which are only possible in such models. Further, these models allow investigation into specific domains of behaviour and cognition with high face validity [9,10], ultimately improving our understanding of the mechanisms underpinning these risks and the genesis of subsequent behavioural phenotypes with a greater degree of granularity than clinical research permits. One important advantage is the ability to investigate behaviour along a developmental timeline. As cognitive deficits can be seen prior to the appearance of “classical” symptoms in schizophrenia (e.g., psychosis) [5], the use of preclinical models allows investigation into the pathophysiological mechanisms that may drive these deficits, independent of other contributing factors such as socioeconomic status [11], and can track the emergence of such deficits within the framework of advancing development.

MIA models vary in regards to species, timing and method of immune activation, with the bacterial cell-wall component lipopolysaccharide (LPS) and the double-stranded RNA mimetic polyinosinic:polycytidylic acid (poly(I:C)) being the most common immunostimulants used [9,12]. These methods result in innate immune system activation via toll-like receptor (TLR) 4 and 3 pathways, respectively [13]. Ultimately, this can result in a foetal inflammatory response that differs in temporality and magnitude [9].

We have previously validated a rat model of MIA [14,15] using a single intraperitoneal (i.p.) injection of 10 mg/kg bodyweight poly(I:C) (InvivoGen) on gestational day (GD) 15 in pregnant Wistar rats and demonstrated that this induces an acute inflammatory response through elevated maternal plasma proinflammatory cytokines, IL-6 and TNFα, 3 h post-exposure [14,16]. Further, we have shown that this provokes a robust cognitive deficit in the attentional set-shifting task (ASST) in adult female offspring [17], a phenotype representative of prefrontal cortex (PFC) executive function deficits, observed frequently in schizophrenia patients. Our current work aims to explore the pathological molecular mechanisms which predispose adult offspring to cognitive deficits, with a particular focus on epigenetic mechanisms.

Epigenetic mechanisms, including DNA modifications, histone modifications and non-coding RNA, work together to regulate gene expression in sex-, cell- and tissue-specific patterns [18]. For example, DNA methylation, one of the best-studied epigenetic mechanisms, occurs primarily at cytosine–guanine dinucleotides (CpG sites) within the DNA sequence; when this occurs within gene regulatory elements, such as promoters and CpG islands (CGIs), it leads to direct alteration of gene expression [19,20,21]. Importantly, epigenetic mechanisms can be modulated by environmental exposures and experiences [22,23]. Thus, one of our key hypotheses is that MIA induces epigenetic changes in the foetal brain which in turn promote maladaptive transcriptional programs, resulting in a perturbed developmental trajectory and predisposition to later-life disease. This hypothesis stems from the developmental origins of health and disease research field, which posits that prenatal and early life environments cause maladaptive epigenetic changes which alter normal development [24]. In humans, and particularly in the human brain, this is difficult to study. The study of epigenetic patterns within the brain can only be achieved through live biopsy or postmortem studies, and thus human neuro-epigenetic research is still fraught with methodological and practical challenges [25]. While alternative studies have used epigenetic patterns in blood samples as a proxy for the brain, it is known that these do not always reliably correlate [26]. Nonetheless, studies in neuro-epigenetics can enable the identification of long-term dysregulated cellular pathways, providing the chance to identify novel targets for clinical interventions. In schizophrenia, epigenetic studies using both postmortem brain and blood samples have produced variable results [27]. One of the more robust and most extensively studied epigenetic targets is the *RELN* gene, which encodes the protein Reelin, involved in the regulation of neuronal migration, dendritic growth and branching, synaptogenesis and synaptic plasticity [28]. Indeed, in addition to *RELN* DNA hypermethylation, up to 50% downregulation of Reelin expression has been frequently identified in the cortex and hippocampus of schizophrenia patients [27,29,30].

Reelin is a large extracellular secreted glycoprotein with important roles in neurodevelopment, as well as maintenance of the adult brain, through binding to its dimeric receptors, very low density lipoprotein receptor (VLDLR) and apolipoprotein E receptor 2 (ApoER2), with Reelin binding leading to receptor-mediated endocytosis [28,31]. The role of the Reelin signalling pathway shows developmental specificity, with a distinct functional switch between early and late neurodevelopment. In prenatal development and early postnatal development, Reelin works to regulate neuronal migration and initiation of dendrite outgrowth. For these roles, Reelin is primarily secreted from Cajal–Retzuis cells [28,32], binds to its receptors and promotes subsequent phosphorylation of the cytosolic receptor-associated adaptor protein, Disabled 1 (DAB1), by Src-family kinases [31,32,33,34] (Figure 1A). Phosphorylation of DAB1 leads to the activation of CRK adaptor proteins which activate various downstream signalling pathways, including those which mediate cell adhesion molecules, for cortical neuron migration, and protein translation, for dendrite outgrowth [28,33,34,35] (Figure 1A). In later postnatal development, Reelin secretion becomes restricted to GABAergic interneurons in the cortex and hippocampus, where the function of the Reelin signalling pathway becomes associated with long-term potentiation (LTP) and synaptic plasticity [28], critical for higher cognitive functions. This is achieved through Reelin-receptor binding leading to PSD-95-mediated regulation of glutamatergic N-methyl-D-aspartate (NMDA) receptors and subsequent activation of the CAMKII kinase to promote transcriptional upregulation of proteins required for learning and memory, through ERK kinase activation [36,37,38,39] (Figure 1B). It has also been shown that this can be achieved through non-canonical DAB1 signalling, either through DAB1-mediated PSD-95 activation or via DAB1-mediated AKT-ERK activation [32,38] (Figure 1B). However, the mechanisms regulating these developmental functional switches remain unclear.

Supporting a functional role for observed alterations in the Reelin pathway in schizophrenia, mice harbouring heterozygous mutations in the *Reln* gene (heterozygous *Reeler* mice; HRM) exhibit multiple behavioural abnormalities similar to those seen in schizophrenia, including executive function and reversal learning deficits [40,41,42]. Further, mice with mutations in the signal transduction machinery, including receptors, ApoER2 and VLDLR [43,44], and DAB1 [32,45], possess phenotypic deficits comparable to those observed in *Reeler* mice, including defects in LTP and cognition. Taken together, these findings suggest that developmental Reelin signalling abnormalities can promote the development of cognitive deficits attributed to schizophrenia. However, despite the evident interest of the Reelin signalling pathway in schizophrenia pathophysiology, very little work has been done to evaluate the Reelin pathway in MIA models, and the work that has been completed offers conflicting results [9]. In the studies which have identified significant changes, reduced Reelin-positive cells were identified in the early postnatal cortex, [46] and reduced *Reln* expression was identified in the developing hippocampus [47]. These studies begin to identify a role for Reelin signalling following MIA, further supported by a recent study which demonstrated that injecting Reelin directly into the hippocamps rescued MIA-induced cognitive deficits in affected offspring [48]. While several signalling pathways have shown MIA-induced changes [9], the aforementioned studies provide particular evidence for defective Reelin signalling, following MIA, in the development of cognitive deficits. That said, no studies in MIA models have evaluated Reelin expression across cortical development, and, with the exception of PSD-95, none have evaluated downstream components of this pathway [9]. Further, no studies to date have evaluated epigenetic changes in the Reelin pathway in a MIA model [9], despite this finding being evidenced in schizophrenia postmortem studies [27,29,30]. Hence, following our previous work, which validated a PFC-mediated cognitive deficit in adult female rats exposed to MIA [17], as well as sex-independent deficits in recognition memory [49], our aim here was to evaluate the possible role of the Reelin pathway in contributing to these cognitive deficits. Given the wide range of functional roles for Reelin across neurodevelopment, we sought to explore the pathway across both early and late neurodevelopment to determine if, and when, pathological changes occur. We expanded our behavioural phenotyping tasks by including the PFC-mediated object-in-place (OPT) and hippocampal-mediated object-location tasks in adolescence [50,51] to further understand the time course, magnitude and type of cognitive deficits. Typical performance in these tasks depends on reciprocal connections between the mPFC and hippocampus, with NMDA receptor function in both being necessary for appropriate object-location associations [52]. As Reelin and its downstream markers play a critical modulatory role in the function of NMDA receptors (and subsequently LTP), we hypothesised that we would observe developmental dysregulation of the Reelin pathway, preceding the onset of cognitive deficits.

## 2. Materials and Methods

### 2.1. Animals and Experimental Design

All animal procedures adhered to the Animals in Scientific Procedures Act (ASPA) 1986 and were approved by the Animal Welfare and Ethical Review Body (AWERB) at the University of Manchester. All animal work was conducted at the Biological Services Facility (BSF) at the University of Manchester as described previously [14,15,16,17,49]. Briefly, adult Wistar rats were obtained from Charles River Laboratories and housed in single-sex groups within individually ventilated cages (IVCs, Double-Decker Cage, Tecniplast, Buguggiate, Italy) under standard conditions: 12 h light/dark cycles (07:00–19:00/19:00–07:00), 21 ± 2 °C, 55 ± 5% humidity, with ad libitum access to food and water. While the use of IVCs has been shown to increase spontaneous abortion rate and attenuate offspring behavioural effects in GD9 mouse models of poly(I:C)-induced MIA, induction on GD12 using IVCs resulted in no increase in MIA-induced spontaneous abortion, but still induced offspring deficits in working memory, sociability and mean PPI, akin to open-cage-housed GD9 models [53]. GD12 in mice reflects a developmental period closer to GD15 in rats [9], illustrating a comparable timing to our established model [14,15,16,17,49]. Further, previous analyses within our group have identified no effects on maternal rearing behaviour or offspring cognition from the use of IVCs [54].

Female rats were time-mated with one male, and evidence of a vaginal plug was taken as indication of successful mating (GD1). Pregnant dams were subsequently pair-housed until GD15. On GD15, dams were pseudo-randomly assigned to receive a single 10 mg/kg bodyweight intraperitoneal (i.p.) injection of low molecular weight (LMW) polyinosinic:polycytidylic acid (Poly(I:C); InvivoGen (San Diego, CA, USA); diluted in physiological saline (0.9% NaCl) to a concentration of 10 mg/mL) or physiological saline only (vehicle control). Experimenters were blinded to treatment and injections were performed between 09:00 and 10:00 to minimise the influence of circadian rhythms. On GD16, dams were individually housed with plentiful access to nesting material and randomly assigned to prenatal or postnatal cohorts. 

Dams assigned to prenatal harvests (GD15 and 21) were anaesthetised with 5 L/min isoflurane, 6 L/min NO_2_ and 2 L/min O_2_, until cessation of breathing. Death was confirmed by cardiac puncture and removal of the heart. Foetuses were harvested from the uterine horn and rapidly decapitated, and the brains were collected for further processing. Tail clips were collected to determine foetal sex as described previously [16]. Dams assigned to postnatal cohorts continued to full term (GD23) with birth confirmed by presence of pups in the nest and considered postnatal day (PD) 0. On PD1, pups were sexed based on anogenital distance, marked with an identifiable pattern using a non-toxic marker as described previously [17] for identification and randomly assigned to cull timepoints (pre-weaning: PD1, PD21; pre-behavioural adolescents: PD35; post-behavioural adults: PD100). Pups assigned to post-weaning behavioural testing and harvests were weaned at PD28 and randomly assigned to female (n = 5/cage) and male (n = 2–3/cage) cages following local regulations, with behavioural testing performed from PD50 onwards. All cages contained a small cardboard tube for nesting and nesting material (Sizzlenest) in equal amounts across conditions.

It is important to consider the impact that these items and differences in cage size may impart. Increases in cagemate numbers have been shown not to affect object location in male Wistar rats, as tested currently [55]. With regards to MIA, while environmental enrichment, including two-level caging, increased cagemate numbers and novel item presence, appears to mediate LPS-induced MIA effects on female offspring cognition, increases in cagemate number alone do not [56]. This is in contrast to what is seen in male MIA offspring from the same group [57]. It is important to acknowledge that social enrichment alone interacted with MIA effects, affecting the expression of stress-response and synaptic plasticity-related genes in females only [56]. In this context, it is important to appreciate the role that housing may play in any expression changes seen presently. Despite these considerations, and much like the use of IVCs, this methodology has imparted significant effects of MIA on offspring cognition in our group [17,49]. 

Pups selected for PD1 harvests were removed from the nest and sacrificed by anaesthetic overdose with 5 L/min isoflurane in 2 L/min O_2._ Each pup was then rapidly decapitated and checked for cessation of bleeding. PD1 brains were then removed for downstream analysis. Pups assigned to PD21, PD35 and PD100 culls were removed from the nest and sacrificed with 2 L/min CO_2_ until cessation of breathing, followed by cervical dislocation and removal of brains for further processing. 

For a visual representation of the protocol, see Appendix A. 

### 2.2. Tissue Processing

Brains harvested from sacrificed foetuses and postnatal pups were flash frozen on dry ice and stored at −80 °C for later dissection. With the exception of GD15, brains were dissected frozen on ice: first bisected into hemispheres before isolating the frontal cortex (FC; GD21 and PD1) and prefrontal cortex (PFC; PD21–100), following the Rat Brain Atlas [58]. The FC was isolated only at early time points (GD21 and PD1) where the PFC has yet to fully develop [59]. Dissected left brain tissue samples were stored at −80 °C in 200 μL RNAlater stabilization solution (Sigma-Aldrich, Manchester, UK) for nucleic acid isolation, while matched right hemispheres were flash frozen for protein isolation. 

#### 2.2.1. Nucleic Acid Isolation

For nucleic acid isolation, both genomic DNA (gDNA) and total RNA were extracted from RNAlater-stored frozen tissue samples using the innuPREP DNA/RNA mini kit (Analytik-Jena, Jena, Germany) according to the manufacturer’s instructions. Nucleic acid concentration and quality were assessed using a Nanodrop (Thermo Scientific, Oxford, UK). 

#### 2.2.2. Protein Isolation

One hundred fifty microliters of homogenization buffer (10 mM Trizma base, 2.5 mM Na2EDTA, 0.3 mM sucrose, 1 mM PMSF, 1 mM sodium orthovanadate (Sigma-Aldrich, Manchester, UK) and 1X cOmplete Mini Protease Inhibitor Cocktail (Roche, Basel, Switzerland)) was added to frozen samples. Samples were homogenized on ice using a pestle and, once fully lysed, centrifuged for 15 min (800× *g*, 4 °C; Eppendorf Centrifuge 5402) to pellet cellular debris. The supernatant was then transferred to a fresh Eppendorf tube and centrifuged for a further 20 min (12,000× *g*, 4 °C; Eppendorf Centrifuge 5402) to pellet synaptic membranes. The final supernatant, containing the cytosolic protein fraction, was quantified using a Bradford Protein Assay (Bio-Rad, Watford, UK), according to the manufacturer’s instructions. Protein samples were stored at −80 °C.

### 2.3. Gene Expression Analysis by RT-qPCR

RNA was reverse transcribed into complementary DNA (cDNA) using the QuantiTect Reverse Transcription Kit (Qiagen, Manchester, UK), according to the manufacturer’s instructions, using 800 ng RNA per reaction. cDNA samples were diluted 1:50 in nuclease-free water prior to qPCR reactions. The QuantiFast SYBR Green qPCR protocol (Qiagen, Manchester, UK) was used to determine relative mRNA expression, in combination with pre-optimised QuantiTect primer assays (Qiagen, Manchester, UK) for all candidate genes of interest (*Dlg4*, QT00183414; *Dab1*, QT00188517; *Reln*, QT00195699; *Camk2b*, QT00183407). On each 96-well plate, an eight-point standard curve was prepared by a 2-fold serial dilution using a pooled cDNA sample (1 μL pooled from each cDNA sample, 40 ng/μL). All standards, samples and negative controls were assessed in duplicate. qPCR reactions were performed on the AriaMx Real-time PCR cycler (Agilent, Cheadle, UK) as per kit instructions, with the addition of a machine-default melt curve protocol. Sample cycle threshold (Ct) values were used to interpolate sample concentration (ng/well) from the log-linear plot of the standard curve. All standard curves adhered to the following acceptance criteria: efficiency 90–110%, r^2^ > 0.98 and %CV values < 1 between duplicates. Three stable reference genes, as determined by geNorm analysis (qbase+, Biogazelle. Oost-Vlaanderen, Belgium) [60], were also quantified for each sample*: Ubc*, *Gapdh* and *B2m* (Primer Design Ltd., Southampton, UK). Candidate gene values were subsequently normalised to the geometric mean of the three reference genes. 

### 2.4. DNA Methylation Analysis by Pyrosequencing

#### 2.4.1. Assay Design

PCR and sequencing primers for the *Dab1* promoter (Chr5:127881334–127881382, Rat genome version Rn5.0) and *Camk2b* internal CpG island (Chr14:86896016–86896037, Rat genome version Rn5.0) were designed using the Pyromark assay design software (v2.0., Qiagen, Manchester, UK) and the top-scoring assay, reflecting the confidence of the assay, selected for use (Table 1). Note that an efficient pyrosequencing assay for the *Reln* and *Dlg4* promoters could not be effectively designed due to high CpG densities within the promoter sequences. 

#### 2.4.2. Bisulphite Treatment and Bisulphite PCR

The EZ DNA Methylation-Gold Kit (Zymo research, Irvine, CA, USA) was used to bisulphite treat 100 ng gDNA according to manufacturer’s instructions. Bisulphite-treated gDNA (bisDNA) was eluted in 11 μL volume. A 2 μL bisDNA sample was amplified using a PyroMark PCR Kit (Qiagen, Manchester, UK) according to the manufacturer’s instructions. All bisulphite PCR reactions were performed on a ^3^Primer thermal cycler (Techne, London, UK), following the kit instructions. A positive control bisulphite PCR reaction was performed using a 2 μL bisDNA pool (prepared from 1 μL bisDNA pooled from each sample) while a negative control bisulphite PCR was also performed using only water. Following PCR, amplicon products were checked for correct amplicon size and specificity, using agarose gel electrophoresis. Agarose gels (2% agarose, 0.01% Gel Red stain in 1xTAE buffer) were loaded with 5 μL bisulphite PCR products and 25 bp Hyperladder (Bioline, London, UK) as a molecular weight marker for assessing amplicon size (Appendix A). 

#### 2.4.3. Pyrosequencing

The Pyromark Q24 Advanced Reagent kit (Qiagen, Manchester, UK) was used for preparation of pyrosequencing reactions following the manufacturer’s instructions, using the PyroMark vacuum workstation and PyroMark Q24 sequencer (Qiagen, Manchester, UK). One positive control and one negative control were included on each plate to act as inter-plate controls. The designed assay sequences for analysis (Table 1) were programmed into the PyroMark Q24 software (v2.0., Qiagen, Manchester, UK), and the assays were run according to machine-default settings. Output data included a calculated methylation percentage at each CpG site in the sequences analysed, with the average methylation of all sites being calculated for statistical analysis of a given candidate gene. 

### 2.5. Western Blotting 

Western blotting was used to quantify DAB1 protein expression from cortex lysates. The anti-DAB1 antibody (rabbit monoclonal, ab68461, Abcam, Cambridge, UK) was validated for its specificity and suitability for detecting cytosolic DAB1 across development (Appendix A). Protein and antibody inputs were also optimised prior to analysis (Appendix A) with a protein loading of 50 μg (GD21, PD1) or 75 μg (PD21–100) and a 1:1000 antibody dilution determined to be optimal. All Western blots were performed using the Mini-PROTEAN Tetra Cell Electrophoresis unit, Mini Trans-Blot Module and PowerPac Basic Power Supply (Bio-rad, Watford, UK) using precast Mini-PROTEAN Tris-Glycine eXtended (TGX) Gels (4–15% polyacrylamide, 50 µL 10 wells; Bio-rad, Watford, UK) with 5µL Precision Plus Protein All Blue Prestained Protein Standards (10–250 kDa, 1610373; Bio-rad, Watford, UK) for molecular weight analysis. Secondary IRDye antibodies were used for infrared detection (Li-Cor, Cambridge, UK), including the following: IRDye 800CW Donkey anti-Rabbit IgG (H + L) and IRDye 680RD Donkey anti-Mouse IgG (H + L). 

Briefly, cytosolic lysates samples were heated under reducing buffer conditions (1XLaemmli sample buffer (Bio-Rad, Watford, UK), containing 5% mercaptoethanol (Sigma-Aldrich, Manchester, UK) heated for 5 min at 95 °C). Following electrophoresis, proteins were transferred to Immobilon-FL PVDF membranes (Merck, Darmstadt, Germany) before being blocked (5% (*w*/*v*) Milk Powder (Marvel) in PBS) for 1 h at room temperature. DAB1 primary antibody was diluted 1:1000 in antibody diluent (5% (*w*/*v*) Milk Powder (Marvel, Dublin, Ireland) in PBS/0.05% Tween20 (Sigma-Aldrich, Manchester, UK)) and incubated overnight at 4 °C. Membranes were then washed in PBS/ 0.1% Tween20 prior to probing with reference protein antibody (anti-GAPDH, rabbit monoclonal, 60004-1-Ig; Proteintech, Manchester, UK) diluted 1:2000 in antibody diluent, and incubated for 2 h at room temperature. Membranes were washed again and then probed with secondary IRDye antibodies. The donkey anti-rabbit IRDye 800CW (green) was diluted 1:20,000 and the donkey anti-mouse 680D (red) secondary antibody was diluted 1:40,000, in LiCor secondary solution (0.2% Tween20, 0.01% SDS (Sigma-Aldrich, Manchester, UK) in PBS) and incubated for 1.5 h at room temperature. Membranes were imaged immediately using the LiCOr Odyssey CLX imaging system on default settings. Gels were analysed using the Li-Cor Image Studio (v5.0) software (Li-Cor, Cambridge, UK), with signal intensity from the DAB1 band normalised to signal intensity from the GAPDH reference band within each sample lane. A representative Western blot can be found in Appendix A.

### 2.6. Behavioural Testing

#### 2.6.1. Object Location 

On PD 51, offspring spatial recognition memory was tested using the object-location task (OLT). In brief, animals were placed in a 52 × 52 × 31 cm polyvinyl arena. Cameras were suspended above the centre of the arena to record behaviour. Each arena contained two identical items placed adjacent to one another (Appendix A). Offspring were placed into the centre of the arena and allowed to explore freely for 180 s. Animals were then removed and placed back into their home cage for 300 s. Arenas were cleaned with 70% EtOH to remove olfactory cues, and duplicate items were placed in a novel configuration (Appendix A). Offspring were placed back into the centre of the arena and allowed to explore both items freely for 180 s prior to the conclusion of the test. Videos were recorded and subsequently analysed using BORIS software (v6.2.4, Torino, Italy). [61]. 

Object-location memory has been shown to be both affected by early life and prenatal stress [62,63] and weakly related to hippocampal–prefrontal cross-frequency coupling [62]. Patients have shown deficits in spatial working or recognition memory, which may underpin more complex item-location-binding deficits [64]. The use of the OLT relies on the rats’ natural preference for novelty much like the novel object recognition task [65]. Deficits in the novel object recognition task have been shown to occur in adolescence following poly(I:C)-induced MIA in our group previously [66]. To standardize preference for novel items over familiar ones, a discrimination index was calculated, ranging from −1 (complete preference for familiar objects) to +1 (complete preference for novel objects). A discrimination index of 0 represents no item preference.
$$Discrimination\;Index=\frac{\left(\sum \mathrm{Exploration}\;\mathrm{of}\;\mathrm{Novel}\;\mathrm{Item}\;\left(s\right)\right)-\left(\sum \mathrm{Exploration}\;\mathrm{of}\;\mathrm{Familiar}\;\mathrm{Item}\;\left(s\right)\right)}{\left(\sum \mathrm{Exploration}\;\mathrm{of}\;\mathrm{All}\;\mathrm{Items}\;\left(s\right)\right)}$$

#### 2.6.2. Object in Place (OPT) 

Twenty-four hours following the OLT, offspring were again placed into the centre of the arena, which contained four items: a black ceramic pot (12.5 × 5.0 cm), a brown ramekin, an opaque white bottle (12.5 × 6.5 cm), and either a small transparent brown bottle (12.5 × 6.5 cm) or a Diet Coke can (12.5 × 6.5 cm), whichever was not used as the item in the OLT as to avoid familiarity effects. Animals were allowed to freely explore all four objects for 180 s before being removed and placed back in their home cage for 300 s. Objects were removed and arenas were cleaned with 70% EtOH to remove olfactory cues. Copies of the items were placed back into the arena with the location of two swapped (Appendix A). Animals were placed back into the arena and allowed to freely explore for another 180 s. Videos were recorded and subsequently analysed using BORIS software (v6.2.4, Torino, Italy). [61]. 

The OPT is used to measure associative recognition memory in that it requires recognition of both item identity and location. This task has been shown to be disrupted by prenatal poly(I:C) exposure [50], and recruitment of the mPFC is necessary for the successful integration of object recognition and location information, likely due to its connections to both the perirhinal cortex and hippocampus [51]. The task was ultimately selected to further characterize the prefrontal dysfunction illustrated within our model [17]. Binding of item-location associations has been shown to be disrupted in patients with schizophrenia [65], with patients less accurately recalling item-location changes than healthy controls [67]. As described above, a discrimination index was calculated in a similar manner. 

### 2.7. Statistical Analysis and Sample Size

Sample sizes were determined using the statistical package G*power (v3.1.9.2, G*Power, Germany). based on previous in-house data [16,17,49], which indicate a medium to large effect size (f = 0.25–0.4), Accordingly, a sample size of n = 5–6 and n = 14, per sex and treatment group, for molecular and behavioural analysis, respectively, was calculated for statistical power (1 − β) = 0.8, with a type I error rate (α) = 0.05. 

All statistical analyses were performed using SPSS statistics (v28, IBM, Chicago, IL, USA) with general linear mixed models (GLMMs) used to analyse offspring traits. For molecular analyses, dam was used as a random factor, with treatment and sex as fixed factors. A minimal model was used for all molecular analyses to determine the minimal number of statistically significant factors within a model. For behaviour analysis, models were parameterized to include two-way between-subject interactions. Discrimination indices for the OLT and OPT tasks were analysed using a generalized linear mixed model (GLMM) with the dam included as a random factor where necessary. Validation of novelty preference in the OPT and OLT used one-sample t-tests. Effects of item (OPT) were analysed by post hoc pairwise comparison with Bonferroni corrections. Where model assumptions were violated in repeated measure generalized linear models (RM-GLMs), Greenhouse–Geisser estimates were applied. Satterthwaite’s estimate was applied where appropriate in general linear models.

All graphs were produced using GraphPad Prism (v9.4, GraphPad Prism Inc., San Diego, CA, USA). Data are presented as mean ± standard error of the mean (SEM), with dam (N) and offspring (n) n-numbers provided in the figure legends.

## 3. Results

Our first aim was to identify cognitive deficits, specifically in spatial recognition, akin to those seen in schizophrenia patients [64]. We were especially interested in an adolescent/early adult age to identify when the cognitive deficit phenotype emerged, as previous research in our group had identified prefrontal dysfunction in adulthood [17]. The tasks (OLT and OPT) were selected due to their translational and ethological validity [10,65] as well as their potential relevance to regions affected in schizophrenia [62].

### 3.1. MIA Selectively Affects Spatial Object Memory

#### 3.1.1. Object-Location Task

In the acquisition phase, adolescent offspring exhibited no preference for items based on location (RM-GLM: F_1,56_ = 0.361, *p* = 0.550), though males spent less time exploring the items at a trend level (GLMM: F_1,52.189_ = 3.460, *p* = 0.069, Figure 2A). In the retention phase, total exploratory time in the task was not affected by MIA (GLMM: F_1,4.006_ = 0.310, *p* = 0.607), but item exploration was significantly lower in males than in females (GLMM: F_1,52.297_ = 7.933, *p* = 0.007). The interaction between the sex and group was not significant (GLMM: F_1,52.297_ = 0.001, *p* = 0.977). Vehicle offspring exhibited a significant preference for the novel object (t(30) = 2.625, *p* = 0.014). In contrast, MIA offspring exhibited no preference (t(29) = −0.785, *p* = 0.439), suggesting a deficit in visuospatial recognition memory. Discrimination index in the OLT was significantly affected by MIA (GLMM: F_1,56_ = 6.069, *p* = 0.017) with no sex effects present (GLMM: F_1,56_ = 0.140, *p* = 0.710), though a trend toward an interaction between MIA and sex is noted (GLMM: F_1,56_ = 3.739, *p* = 0.058, Figure 2C).

#### 3.1.2. Object-in-Place Task

In the acquisition phase, there were no effects of MIA (GLMM: F_1,3.470_ = 0.285, *p* = 0.626), sex (GLMM: F_1,48.535_ = 0.775, *p* = 0.383) or an interaction between these two factors (GLMM: F_1,48.535_ = 0.451, *p* = 0.505) on total exploration time in adolescent offspring. This was also true in the retention phase where neither MIA (FLMM, F_1,3.551_ = 0.300, *p* = 0.616), sex (GLMM: F_1,48.518_ = 2.908, *p* = 0.095) nor an interaction between the two significantly affected exploration time (GLMM: F_1,48.518_ = 0.001, *p* = 0.981). In both the acquisition and retention phase, offspring exhibited an object-dependent preference (Acquisition: RM-GLM: F_2.254, 117.210_ = 17.412, *p* < 0.001; Retention: RM-GLM: F_2.128, 110.631_ = 52.106, *p* < 0.001, Table 2; Figure 3A–C).

In terms of discrimination index, no difference between vehicle and MIA offspring was seen (GLMM: F_1,52_ = 2.901, *p* = 0.095, Figure 3C), and neither sex (GLMM: F_1,52_ = 0.448, *p* = 0.506) nor an interaction between the two factors was significant (GLMM: F_1,52_ = 0.019, *p* = 0.890). However, vehicle offspring exhibited a trend towards a novel object preference (t(27) = 1.965, *p* = 0.060, Figure 3C), whereas MIA offspring exhibited no preference (t(27) = −0.316, *p* = 0.755).

### 3.2. MIA Induces Reductions in Reln Gene Expression Prenatally and in the Adult Prefrontal Cortex

*Reln* was significantly downregulated 3 h post-treatment in the foetal whole brain in both sexes (GLMM: F_1,20_ = 78.92, *p* < 0.001; Figure 4A), suggesting an acute and rapid dysregulation of this pathway in response to MIA. We next assessed the longitudinal expression of *Reln* in the developing cortex from prenatal to postnatal timepoints. In line with previous literature [28,68], we found in our study an overall higher *Reln* gene expression in prenatal and early development, peaking at PD1. We found that in the GD21 FC, there was a significant sex*group interaction (GLMM: F_3,16_ = 3.57, *p* = 0.038), with post hoc analysis by sex showing a significant reduction in *Reln* expression in the FC of MIA females relative to vehicle females (GLMM: F_1,8_ = 6.17, *p* = 0.038; Figure 4B), with no significant main effects in the males. These data suggest that the acute reduction in *Reln* expression observed at GD15 persists in a sex-dependent pattern throughout pregnancy. Postnatally, there were no significant main effects until PD35, when we found a trend to a main effect of group (GLMM: F_1,12.80_ = 3.93, *p* = 0.069) and a trend of a group*sex interaction (GLMM: F_2,11.98_ = 3.41, *p* = 0.067). Post hoc analysis by sex showed a significant increase in *Reln* expression in the PFC of MIA females relative to vehicle females (GLMM: F_1,12.0_ = 4.90, *p* = 0.047; Figure 4B) with no significant main effects in the males. However, by PD100, there was a significant main effect of sex (GLMM: F_1,14.8_ = 5.43, *p* = 0.034; Figure 4B), with males having overall higher *Reln* expression in the PD100 PFC than females, and a main effect of group (GLMM: F_1,15.3_ = 4.45, *p* = 0.050; Figure 4B), with a reduction in *Reln* expression in the PFC of MIA offspring relative to vehicle offspring.

These results support findings in previous MIA models and in schizophrenia postmortem brain studies [29]. However, given that few studies have explored what functional changes to the Reelin signalling pathway exist beyond altered Reelin expression, we next investigated whether there was altered expression of downstream pathway components, in particular the DAB1 adaptor protein, which has also been linked to cognitive deficits similar to those observed in MIA models [32].

### 3.3. Developmental Disturbances in Dab1 mRNA and DAB1 Protein Expression Occur Subsequent to MIA-Induced Reelin Dysregulation

DAB1 serves as the primary downstream adaptor protein for Reelin signalling (Figure 1). Phosphorylated DAB1 works as the active form of this adaptor and is rapidly degraded by ubiquitination to limit prolonged Reelin signalling [28]. Mutations in *Dab1* have been shown to produce similar phenotypes to those observed with Reelin mutants; however, no studies to date have evaluated *Dab1* expression in MIA models. We thus measured *Dab1* mRNA, DAB1 protein and phospho-protein expression throughout development as a measure of functional Reelin signalling.

#### 3.3.1. *Dab1* mRNA

In the FC at GD21, there was a trend to a main effect of group (GLMM: F_1,19_ = 4.16, *p* = 0.056; Figure 5A) with an increase in *Dab1* expression in MIA relative to vehicle control foetuses. Likewise, in the PD1 FC, there was a significant main effect of group (GLMM: F_1,13.96_ = 7.71, *p* = 0.015; Figure 5A) though with a reduction in *Dab1* expression observed in MIA relative to vehicle offspring. Of note, this reversed trend in *Dab1* mRNA expression observed between GD21 and PD1 coincides with a shift in the function of the Reelin signalling pathway from neuronal migration to dendrite formation. Further, in the PD21 PFC, there was again a main effect of group (GLMM: F_1,28_ = 5.22, *p* = 0.030; Figure 5A) with increased *Dab1* expression in MIA offspring relative to vehicle offspring. Again, this coincides with the second functional switch in Reelin signalling from dendrite growth to synaptic plasticity. This elevation in *Dab1* expression remains at PD100, when there was a main effect of group (GLMM: F_1,130_ = 5.93, *p* = 0.021; Figure 5A), alongside a main effect of sex (GLMM: F_1,30_ = 25.05, *p* < 0.001; Figure 5A), where we observed, as with *Reln* expression, higher *Dab1* expression in males relative to females. Taken together, these results suggest ongoing dysregulation in downstream Reelin signalling, occurring along the developmental timeline, during which Reelin signalling undergoes key developmental functional switches.

#### 3.3.2. DAB1 Protein

DAB1 protein stability is primarily regulated by Reelin signalling, with Reelin signalling driving phosphorylation and activation of DAB1, with phospho-DAB1 subsequently degraded to limit Reelin signalling [28]. With this in mind, we next investigated the DAB1 protein:phospho-protein ratio. 

We found no significant differences in DAB1 protein expression prenatally, but there was a significant reduction in DAB1 protein expression in MIA relative to vehicle offspring in the PD1 FC (GLMM: F_1,20.81_ = 12.09, *p* = 0.002; Figure 5B), corresponding to the same finding for *Dab1* gene expression (Figure 5A). Likewise, in the PD21 PFC, there was a trend to an increase in DAB1 protein expression in MIA relative to vehicle offspring (GLMM: F_1,12.20_ = 3.93, *p* = 0.071; Figure 5B) corresponding to a similar trend for *Dab1* gene expression (Figure 5A), alongside a main effect of sex (GLMM: F_1,11.50_ = 5.44, *p* = 0.039; Figure 5B), with increased DAB1 protein expression in females relative to males. Again, these changes in DAB1 protein expression map well to the stages during which Reelin signalling undergoes key developmental functional switches (Figure 1). However, there were no significant changes observed in later development. Next, we evaluated the relationship between *Dab1* gene expression and DAB1 protein expression. We found that there was a positive correlation between the two during early development (GD21-PD21) during which the primary functions of DAB1 are in cortical layering and dendrite maturation (Spearman Rho = 0.689, *p* < 0.001; Figure 5C). However, in later development (PD35–100), there was an inverse correlation between the two (Spearman Rho = −0.301, *p* = 0.021; Figure 5D), when the function of DAB1 would be driven more towards synaptic plasticity. 

Given these changes in DAB1 protein expression, we were next interested in the phospho-form of DAB1 which marks its critical activity. However, despite testing three antibodies from separate suppliers (Cell Signalling Technology, Danvers, MA, USA (3327S); St John’s Laboratory, London, UK (STJ196282); Abcam, Cambridge, UK (ab78200)), no immunoreactive band at the correct size could be detected (Appendix A). Given that phospho-DAB1 is rapidly degraded, we inferred that the active form of DAB1 may be present at too low an abundance in our samples to be detectable by this methodology. Considering this challenge, we instead evaluated further downstream components in the pathway. In particular, given our evidence for an adult cognitive deficit in MIA-affected offspring [17] we were interested in the later developmental Reelin signalling pathway, which contributes to LTP through NMDA receptor modulation [28] (Figure 1B). For this purpose, we chose to examine genes encoding PSD-95 (encoded by *Dlg4*) and CAMKII (encoded primarily by *Camk2a* and *Camk2b* in the brain). 

### 3.4. Changes to PSD-95 and CAMKII Expression Are Observed in the Later Postnatal Period Following MIA 

PSD-95 and CAMKII are both activated downstream of later developmental Reelin signalling, critical for LTP, higher cognitive function and synaptic plasticity, and, hence, this pathway becomes the primary downstream Reelin signalling pathway in early adolescence into adulthood [28] (Figure 1). 

#### 3.4.1. *Dlg4* (PSD-95)

We found no significant differences by group in *Dlg4* gene expression prenatally, although there was a main effect of sex (GLMM: F_1,18_ = 5.29, *p* = 0.034; Figure 6A) with females having overall higher *Dlg4* expression than males. Postnatally, there were only trends to a significant main effect of group at PD1 (GLMM: F_1,20_ = 3.85, *p* = 0.064; Figure 6A) and PD21 (GLMM: F_1,26_ = 3.79, *p* = 0.062; Figure 6A) with MIA offspring showing a reduction in *Dlg4* expression. Of note, this disturbance, as with *Dab1*, corresponds to changes in the function of the Reelin signalling pathway in the early postnatal period. In adulthood, there is a significant reduction in *Dlg4* expression in MIA offspring (GLMM: F_1,27_ = 5.15, *p* = 0.031; Figure 6A). This finding supports a Reelin signalling deficit in adulthood, corresponding to the timeframe of the emergence of a PFC-mediated cognitive deficit [17].

#### 3.4.2. *Camk2b* (CAMKII)

As with *Dlg4*, there were no significant differences in *Camk2b* expression in the prenatal and early developmental stages examined. The first significant main effect is observed in the PD21 PFC, with an increase in *Camk2b* expression in MIA offspring relative to vehicle offspring (GLMM: F_1,24_ = 6.89, *p* = 0.015; Figure 6B). In the PD100 PFC, there was a significant sex*group interaction (GLMM: F_3,25.43_ = 3.71, *p* = 0.024; Figure 6B). Post hoc analyses by sex showed a significant increase in *Camk2b* expression in MIA females relative to vehicle females (GLMM: F_1,13.96_ = 5.05, *p* = 0.041; Figure 6B) and a reduction in MIA males relative to vehicle males (GLMM: F_1,12.0_ = 5.21, *p* = 0.041; Figure 6B). These changes in the expression of *Camk2b* later in development align with the change in function of Reelin signalling to incorporate this marker.

### 3.5. MIA-Induced Changes in Gene Expression Precede Developmental Changes to Dab1 and Camk2b DNA Methylation

To understand the developmental mechanisms which regulate the observed changes in the expression of these pathway components, we investigated aspects of epigenetic regulation. In schizophrenia, hypermethylation of the *RELN* promoter has been identified in several studies, corresponding with reduced *RELN* expression [29]. We therefore investigated promoter and/or CGI methylation in our gene targets.

#### 3.5.1. *Dab1* Promoter Methylation

In the GD21 FC, we found no significant main effect of any predictor variable. However, a post hoc analysis by sex did identify a trend to a significant main effect of group within females (GLMM: F_1,8.0_ = 4.32, *p* = 0.071; Figure 7A), with reduced methylation in the MIA females relative to vehicle females. In the PD1 FC, there was a significant main effect of group (GLMM: F_1,13.36_ = 5.18, *p* = 0.040; Figure 7A) with reduced promoter methylation in MIA offspring relative to vehicle offspring. In later development, there was a trend to a main effect of group at PD35 (GLMM: F_1,23_ = 3.41, *p* = 0.078; Figure 7A), with MIA offspring, once again, having lower promoter methylation than vehicle offspring. However, by PD100, the main effect of group was significant (GLMM: F_1,15.59_ = 10.21, *p* = 0.006; Figure 7A), with reduced promoter methylation in the MIA offspring. 

We next assessed the relationship between *Dab1* promoter methylation and changes in gene and protein expression. At GD21, there were no significant correlations between *Dab1* promoter methylation and gene or protein expression. However, at PD1, there was a significant positive correlation between promoter methylation and Dab1 gene expression (Spearman Rho = 0.605, *p* = 0.004; Figure 7C), which becomes a negative correlation at PD21 (Spearman Rho = −0.449, *p* = 0.036; Figure 7D). At PD35, there was again a significant negative correlation between promoter methylation and *Dab1* expression (Spearman Rho = −0.619, *p* < 0.001; Figure 7E), but also a trend to a positive correlation with DAB1 protein expression (Spearman Rho = 0.362, *p* = 0.075). At PD100, as with PD21 and PD35, there was a significant negative correlation between *Dab1* gene expression and promoter methylation (Spearman Rho = −0.402, *p* = 0.038; Figure 7F). These results imply the relationship between *Dab1* promoter methylation and *Dab1* gene expression is dependent on the stage of development; however, these do not always result in temporally matched changes in DAB1 protein expression.

#### 3.5.2. *Camk2b* CGI Methylation 

For methylation of the CGI in *Camk2b* at GD21, there was a significant main effect of both group (GLMM: F_1,9.24_ = 8.34, *p* = 0.034; with increased CGI methylation in MIA offspring relative to vehicle) and sex (GLMM: F_1,9.24_ = 7.93, *p* = 0.020; with overall increased CGI methylation in females relative to males; Figure 8A). This pattern does not persist from this point, with no significant differences in CGI methylation observed again until PD100, where there was a significant group*sex interaction (GLMM: F_3,18.79_ = 5.16, *p* = 0.009). When the data were subdivided by sex, there was a significant main effect of group in females (GLMM: F_1,10.0_ = 9.70, *p* = 0.011; Figure 8A), with reduced methylation in MIA females compared to vehicle, and a significant main effect of group in males (GLMM: F_1,10.02_ = 6.30, *p* = 0.031; Figure 8A), with increased CGI methylation in the MIA males relative to vehicle. 

As with *Dab1* promoter methylation, we next examined the relationship between DNA methylation and expression. However, there were no significant correlations between CGI methylation and *Camk2b* mRNA expression, other than at PD100 (Figure 8C–E), where there was a negative correlation between *Camk2b* mRNA expression and CGI methylation (Spearman Rho = −0.469, *p* = 0.019; Figure 8F).

## 4. Discussion

Our results show epigenetic, transcriptional and translational alterations in the prefrontal Reelin pathway that occur both pre- and postnatally, significantly earlier than the onset of prefrontal-dependent cognitive deficits as reported by Potter et al. (2023) in our MIA model [17]. Many of these differences are absent during a pubertal period (PD35), after which spatial recognition deficits are identified, highlighting the neurodevelopmental timing of emergent deficits. This timing also aligns with identity recognition memory deficits reported previously [49]. It is possible that the occurrence of these phenotypes links to altered NMDA function due to *Dab1* and *Camk2b* protein or gene expression differences resulting in abnormal LTP or LTD, specifically in the PFC. 

The data presented here expand on our previous findings of visual recognition memory and executive function deficits [17,49], this time in the hippocampus-dependent OLT. This task was selected due to its similarities to other recognition memory tasks such as novel object recognition (NOR) [49] as well as similarities to protocols used in human subjects [64]. Generally, our results align with those seen in these studies [64] and may reflect altered hippocampal–prefrontal interactions [62]. Ragland and colleagues (2017) identified poorer performance and posterior hippocampal recruitment during a spatial recognition task in schizophrenia patients [69], suggesting comparable cognitive deficits to those presented here. In accordance with this work, the results of our OLT have illustrated potential hippocampal cognitive deficits to be investigated in further research, as well as face validity to deficits seen in the clinic. As this task was run only in adolescence, future research should investigate whether deficits persist into adulthood as seen in prefrontal-dependent tasks [17] or if their onset pre-empts puberty, a critical period in the development of psychosis and schizophrenia, and one in which cognitive deficits appear to reach significance [5]. As these deficits have been shown to occur prior to the display of other symptoms [5], the understanding of how these spatial recognition memory deficits may interact or predict subsequent phenotypes is critical to understanding the role of neurodevelopmental mechanisms in the MIA model.

The results for the OPT were less clear, however, with MIA offspring illustrating no preference for the novel objects, while VEH offspring did so only at a trend level. Discrimination indices did not differ significantly, likely owing to large intra-group variability as acknowledged in our previous research [49]. It is also possible that age of testing played a role. Research using adult (PD60–90) Long Evans rats in a poly(I:C) model has shown sex-specific effects, in that female offspring were unable to complete the task or show object preference at all [70], but we note this experiment utilised a significantly longer delay between acquisition and retention testing than in our model. 

The Reelin signalling pathway has been frequently implicated in the pathophysiology of schizophrenia, from both postmortem expression and epigenetic studies in patients [29]. Our behavioural data here, and previously [17,49], have demonstrated robust MIA-induced cognitive deficits in learning and memory, analogous to those observed following in vivo dysregulation of the Reelin signalling pathway and in schizophrenia patients [28,32]. Given these parallels, and the limited number of studies to date evaluating changes in the Reelin pathway in MIA models [9], we evaluated functional and regulatory changes to the Reelin signalling pathway. We have focussed on the PFC, given our frequent identification of robust PFC-mediated cognitive deficits in our adolescent and adult rats. We notably observe longitudinal perturbations in the expression and epigenetic regulation of Reelin signalling components occurring concurrently at developmental timeframes during which Reelin signalling undergoes developmental functional switches (Figure 1). In the prenatal phases (GD15–21), Reelin signalling primarily regulates neuronal migration, while immediately postnatally (PD1), Reelin signalling functions in dendrite outgrowth, and in the later postnatal period (PD21–100), Reelin signalling becomes coupled to NMDA-driven LTP and synaptic plasticity. The observed disturbances in Reelin signalling within each of these discrete developmental periods will have distinct effects on neurodevelopment and behavioural outcomes. 

We have shown reduced *Reln* mRNA expression within the first 3 h post-MIA, persisting in female foetuses until GD21, indicating a rapid and prolonged prenatal deficit in *Reln* expression in response to MIA. Postnatally, we observed no significant differences in *Reln* expression until later development, where an increase in adolescence was followed by a decrease in adulthood. Notably, the latter finding corroborated studies using postmortem PFC samples from schizophrenia patients [29,30]. It is well established that Reelin signalling deficits in adulthood disturb LTP, induce reduced synaptic plasticity and contribute to the development of cognitive phenotypes [36,71]. We thus sought to further validate the consequences of the observed changes in *Reln* expression. We assessed the expression of downstream components of the Reelin signalling pathway, beginning with the DAB1 adaptor protein which appears to, in part, mediate the developmental functional switches in Reelin signalling (Figure 1). We observed no significant change in the expression of either *Dab1* gene or DAB1 protein prenatally. However, there was a marked reduction in both *Dab1* mRNA and DAB1 protein expression in the PD1 FC. Of note, there were no statistically significant differences in the expression of later-developmental LTP-associated genes, *Dlg4* (PSD-95) and *Camk2b* (CAMKII), in this early developmental period. This is not surprising given that these genes are more closely associated with the later developmental function of Reelin signalling (Figure 1B). Nonetheless, the marked reduction in DAB1 would suggest an overall decrease in the capacity for Reelin signalling at this stage of development. This may be a consequence of the reduction in prenatal *Reln* expression, resulting in a prolonged reduction in the ability of early neurons to respond to Reelin signalling in the early postnatal period, perhaps leading to reduced dendrite growth in MIA offspring, comparable to those phenotypes observed in Reelin-deficient mice [72].

At the onset of the critical juvenile-adolescent period (PD21–35), during which LTP and synaptic plasticity work to promote normal learning and memory [73], we observed an induction in the expression of the downstream components of the Reelin signalling pathway in MIA offspring, with increased expression of *Dab1* mRNA, DAB1 protein, *Dlg4* and *Camk2b* mRNA. One might hypothesise that this increase could be compensatory for the reduced Reelin signalling observed in early development (GD15-PD1). However, we propose that this observed increase in Reelin signalling may be maladaptive. Indeed, the co-occurrence of this increase with the second temporal switch in Reelin signalling may result in early onset LTP at the incorrect developmental window and, without the correct synaptic maturation, lead to long-term dysregulation in synaptic plasticity. Indeed, by PD100, we observed reduced *Reln* gene expression, increased *Dab1* and reduced *Dlg4* gene expression and sex-specific dysregulation of *Camk2b* in the MIA offspring. This unparalleled dysregulation of Reelin signalling components across the later developmental window is hypothesised to induce NMDA receptor dysregulation and overall reduced plasticity, thereby predisposing offspring to learning and memory deficits. 

We note that there are key directional discrepancies at later developmental timepoints between *Reln* expression and *Dab1*/DAB1 expression. We hypothesise this is likely driven by the role of Reelin in modulating the stability of DAB1. Indeed, Reelin signalling is known to regulate DAB1 phosphorylation and degradation [26,32], while *Reeler* mice, which have reduced Reelin availability, have increased DAB1 protein expression [74]. Accordingly, when assessing the relationship between *Dab1* gene and DAB1 protein expression, we observe positive correlations between mRNA and protein expression from GD21 to PD21 and a negative correlation from PD35 to PD100. Given that these latter stages of development correspond to the time period when DAB1 protein expression does not show a MIA treatment effect, we suggest this inverse relationship between *Dab1* gene and DAB1 protein expression may result from dysregulated DAB1 protein degradation. Notably, in *Reeler* mice, which display DAB1 overexpression, there is a reduced phospho-DAB1 ratio. However, it was not possible here to quantify the phospho-form in the context of our MIA model, due to the lack of detectable phospho-DAB1, despite the use of three different commercial antibodies.

We lastly considered whether the changes in expression of these pathway components are driven by epigenetic dysregulation, as observed in schizophrenia patients [27]. We assessed DNA methylation at the *Dab1* promoter and *Camk2b* CGI. Interestingly, we have shown that changes in mRNA expression of these genes appear to temporally precede changes in the methylation of the regulatory elements. We therefore propose that MIA-induced dysregulation of the expression of these genes may initially be driven by other epigenetic mechanisms, such as histone modifications or transcription factor binding, rather than DNA methylation. This finding might suggest some plasticity in the regulation of this pathway which would allow for interventions which could re-direct the expression of these genes prior to the establishment of the DNA methylation patterns. 

We noted several cases of sex-specific differences in our molecular data. There is a well-recognised sex-bias in neuroscience research, particularly MIA models, where many studies selectively use male offspring in downstream behavioural and molecular analysis [75]. Some of the sex differences identified in this study were independent of MIA, and likely represent normal temporal differences in gene expression patterns in the developing brain in males and females [76]. However, we noted some key sex-specific molecular changes in response to MIA in utero*,* in line with multiple previous studies in MIA models [9]. These included female-specific reductions and increases in *Reln* mRNA expression at GD21 and PD35, respectively (Figure 4) and a marked opposing sex-difference in CAMKII expression and CGI methylation in adult (PD100) offspring in response to MIA (Figure 6B and Figure 8A). *Reln* is well recognised as a risk-factor gene for schizophrenia [29,30]; however, some *Reln* gene variants have been suggested to increase schizophrenia risk in females only [77,78]. This might in part explain the more exacerbated phenotype for *Reln* deficits in females relative to males. CAMKII functions as a holoenzyme comprising various subunits of *Camk2* gene isoforms [79], and hence future work may be needed to identify if there is further sex-specific dysregulation in isoform modalities, which we have not explored in this study. That said, CAMKII has been well documented to have sex-specific roles and responses during LTP [80,81]. Further, a recent study in a chronic unpredictable stress model showed increased hippocampal CAMKII expression in male mice but reduced CAMKII expression in female mice, consistent with the findings for *Camk2b* expression here in response to MIA [82]. This suggests that MIA and other environmental stressors can induce a sex-specific adaptive function of CAMKII, relevant for the development of cognitive deficits. This is supported further in our study, which showed a significant correlation between *Camk2b* CGI methylation and mRNA expression in PD100 MIA offspring, irrespective of sex. This suggests that both the transcriptional regulation and expression of CAMKII were altered in a sex-specific pattern in response to MIA. Overall, the findings support a degree of sex-specific dysregulation of Reelin signalling in response to MIA, which appears to converge in similar cognitive deficits. In line with these sex-specific changes following MIA, there is evidence of differences in schizophrenia molecular pathophysiology between males and females [77,83]. Such findings advocate the need to study both sexes when attempting to unravel the pathological changes that link MIA to schizophrenia behavioural deficits. 

There are some limitations to this study that require further investigation. First, there is a lack of direct evidence for a causal relationship between MIA-induced changes in the Reelin signalling pathway and alterations in offspring cognition. Moreover, it is not possible to establish the temporal windows during which Reelin signalling deficits contribute to the outcome of PFC cognitive functions. To address these limitations, future studies should aim to employ intervention approaches, in line with previous studies [48], to rescue the evidenced PFC Reelin deficit and therefore establish whether such deficits contribute to the development of cognitive impairments. Intervention techniques would ideally be trialled in the early (~PD35) and late (~PD60) adolescent periods to (i) be clinically translatable to the disease of interest and (ii) coincide with the presentation of behavioural deficits identified in MIA-exposed offspring. Nonetheless, our data provide support for the role of Reelin signalling in MIA-induced behavioural deficits, with particular relevance for neurodevelopmental disorders. 

Despite these limitations, our findings, showing a relationship between MIA-induced dysregulation of the Reelin signalling pathway and the emergence of cognitive deficits, are supported by the literature. Critically, the observed hippocampal and PFC-mediated cognitive deficits align with those demonstrated in recent functional studies. Indeed, genetic reductions in both DAB1 in the neonatal period (PD1–7) [84] and later postnatal reductions in PSD95 (*Dlg4)* [85]*,* aligning with the temporal and directional changes evidenced here, produce similar working and spatial memory deficits to those presented in this study. Further, Reelin supplementation in adulthood, where we have observed a reduction in *Reln* expression, has been shown to rescue visual recognition memory deficits in MIA-exposed offspring [48]. Taken together, these studies support a substantial functional relationship between the observed molecular changes and cognitive deficits shown here, through spatial memory deficits, and previously, through visual recognition [49] and cognitive flexibility [17]. Indeed, the hippocampal–mPFC circuit is needed for effective object-location acquisition in this task, with specific importance placed on NMDA receptor function [52]. It is thus possible that the poor novelty preference in MIA offspring is the result of aberrant NMDA receptor function along this axis, ultimately affecting object-location acquisition (Figure 2A,B). This putative circuit is supported by the changes to *Camk2b* and *Dlg4* expression in juvenile and adult animals (Figure 8A,B) which could affect the capacity for LTP/LTD [37]. Interestingly, an adolescent high-fat diet has been shown to selectively reduce Reelin-positive cells without affecting the downstream DAB1 in the mPFC*,* resulting in deficient NMDA-dependent LTD and pronounced cognitive deficits [85]. While the change to *Reln* during adolescence in our model is sex-specific and directionally opposite to these findings, the lack of changed DAB1 agrees well with our findings in later development. It may be, in this case, that the adolescence represents a period of transition toward reduced Reelin signalling as seen in PD100, aligning with findings from previous research [85]. Of note, the fluctuations we observe in this pathway in the later postnatal period (PD21–100) and in the patterns of DNA methylation suggest a developmental window during which attenuation of this pathway may provide ameliorating benefits. Indeed, Reelin supplementation has been shown to improve cognitive flexibility and task performance in MIA models [48] and other models [65].

## 5. Conclusions

Overall, our data demonstrate that MIA induces epigenetic and functional dysregulation of the Reelin signalling pathway. We propose such changes contribute to the emergence of the observed PFC-mediated cognitive deficits we have identified in this model, and therefore provide preliminary evidence of a mechanistic relationship between MIA and developmentally disturbed Reelin signalling. This work supports a role for Reelin signalling in mediating the outcomes of MIA and provides a possible therapeutic target for future study [86].

## Figures and Tables

**Figure 1 biomolecules-13-00489-f001:**
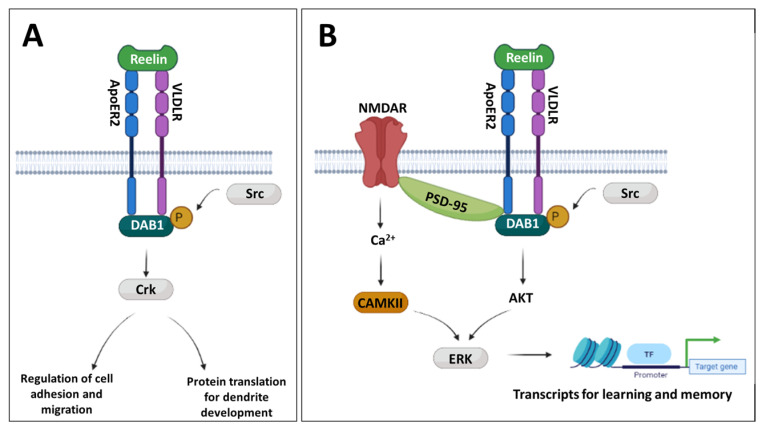
Reelin signalling pathways. (**A**) Overview of prenatal and early developmental Reelin signalling. Reelin binds receptors VDLR and ApoER2, leading to phosphorylation of DAB1 adaptor protein by Src-family kinases. Phosphorylated DAB1 activates the CRK family of proteins, which, in turn, promote signalling cascades which (i) modulate cell adhesion proteins to aid in neuronal migration and lamination or (ii) promote protein translation required for dendrite development. (**B**) Overview of postnatal Reelin signalling. Reelin binds receptors VDLR and ApoER2, leading to (i) activation of PSD-95, which, in turn, activates NMDA receptors (NMDAR), causing calcium influx and activation of CAMKII, and (ii) non-canonical DAB1 signalling and downstream AKT activation, both converging in ERK-driven transcription of genes required for learning and memory.

**Figure 2 biomolecules-13-00489-f002:**
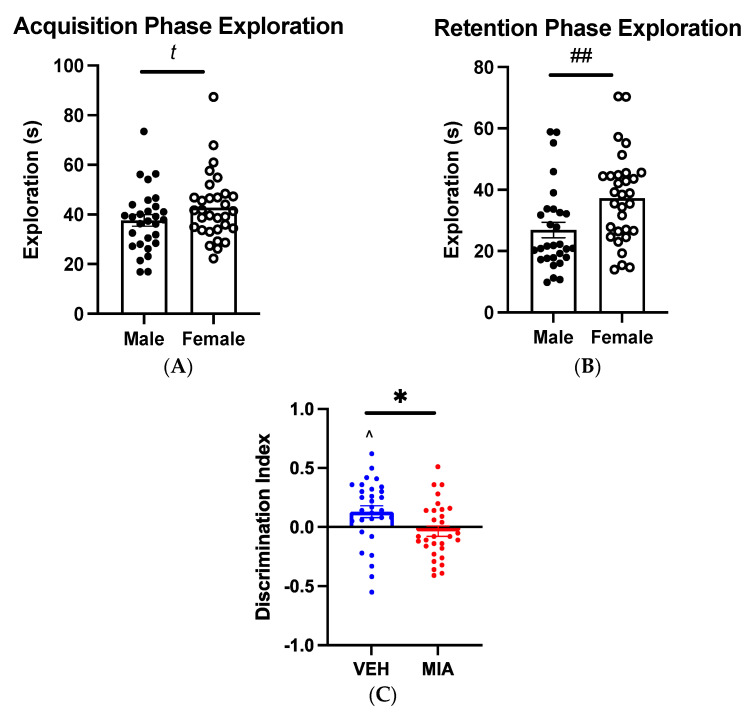
Object−Location Task. Adolescent offspring were tested on the OLT. Individual data points are shown in figure. (**A**) In the acquisition phase, there was a trend toward increased exploration in female offspring, irrespective of prenatal exposure (N = 29–31/sex). (**B**) In the retention phase, female offspring explored objects significantly more than males (N = 29–31/sex). (**C**) Vehicle offspring exhibited a discrimination index significantly different from 0, highlighting a novelty preference. There was a significant difference between the discrimination indices of both groups (N = 30/group). *t p* ≤ 0.075, ## *p* < 0.01 effect of sex, * *p* < 0.05 effect of MIA, ^ *p* < 0.05 difference from 0. Abbreviations: VEH, vehicle; MIA, maternal immune activation.

**Figure 3 biomolecules-13-00489-f003:**
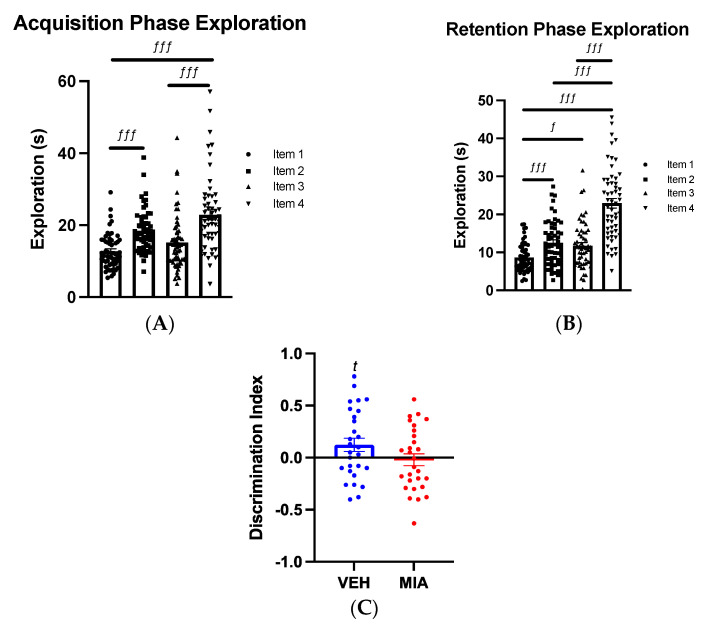
Object−In−Place Task Result. Adolescent offspring were tested on the OPT. Individual data points are shown in figure. (**A**) In the acquisition phase, significant differences in the exploration time of items were found, highlighting a preference for item 4 (n = 60). (**B**) In the retention phase, significant differences in the exploration of items were found, highlighting a preference for item 4 (n = 60). (**C**) Vehicle offspring exhibited a trend towards a novelty preference, whereas MIA-exposed offspring exhibit no such preference (n = 30/group). There was no significant difference between the discrimination indices of both groups. *f = p <* 0.05 item preference, *fff* = *p* < 0.001 item preference; *t* = *p* < 0.075. Abbreviations: VEH, vehicle; MIA, maternal immune activation.

**Figure 4 biomolecules-13-00489-f004:**
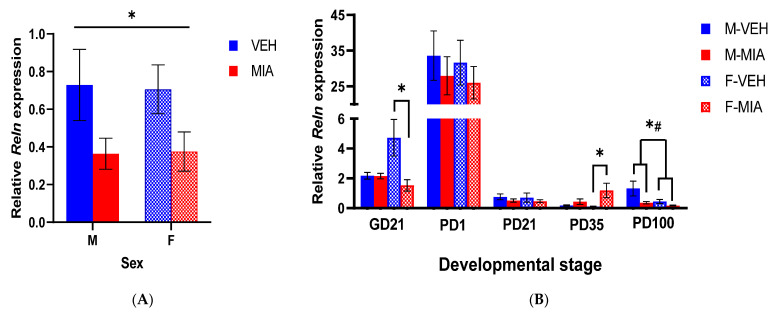
*Reln* gene expression across development. Relative *Reln* mRNA expression, calculated by normalising *Reln* mRNA to the geometric mean of three stable reference genes (*Gapdh*, *Ubc*, *B2m*). (**A**) GD15 whole brain; (**B**) GD21-PD100 cortex. Bars represent mean ± SEM. Results of the GLMM shown (N = 5–7; n = 5–9): effect of group, * *p* < 0.05; effect of sex, # *p* < 0.05; abbreviations: VEH, vehicle, MIA, maternal immune activation; M, male; F, female; GD, gestational day; PD, postnatal day.

**Figure 5 biomolecules-13-00489-f005:**
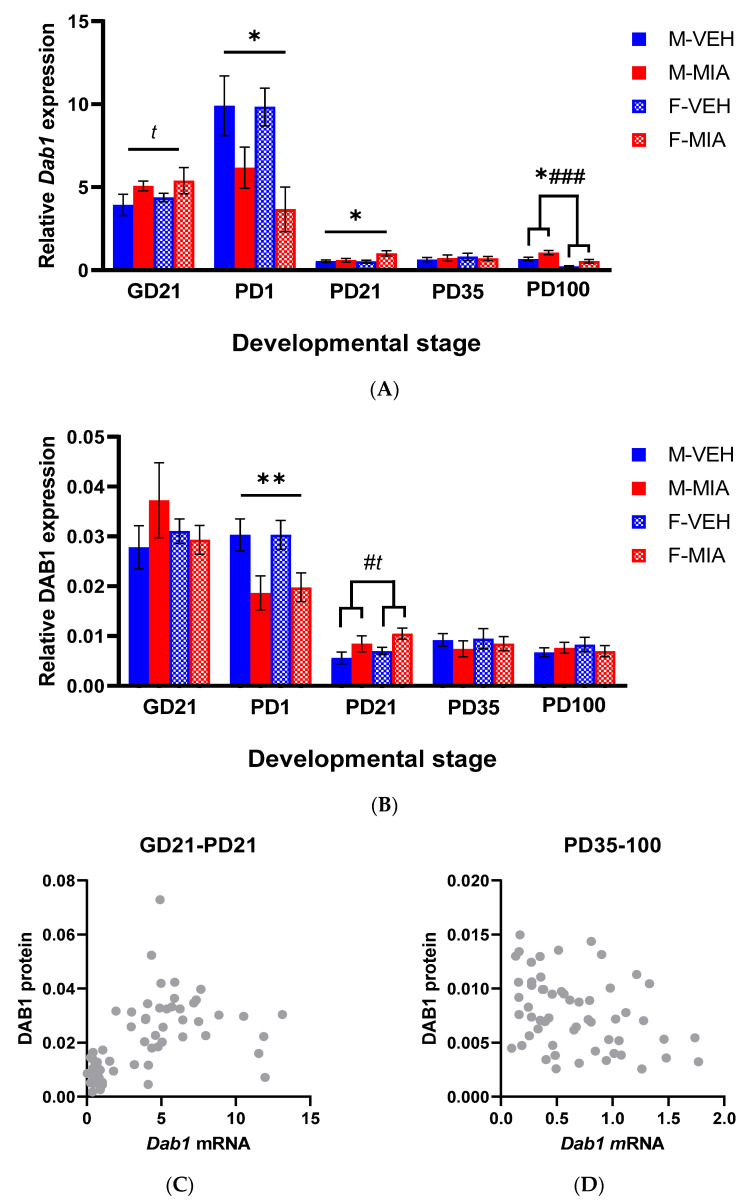
Developmental *Dab1* gene and protein expression. (**A**) Relative *Dab1* gene expression, calculated by normalising *Dab1* mRNA expression to the geometric mean of three stable reference genes (*Gapdh, Ubc, B2m*). (**B**) Relative DAB1 protein expression, calculated by normalising DAB1 protein expression to a stable reference protein (GAPDH). Bars represent mean ± SEM. Results of the GLMM shown (N = 5–7; n = 5–9), trending effect of group, *t p* < 0.075; effect of group * *p* < 0.05, ** *p* < 0.001; effect of sex, # *p* < 0.05, ### *p* < 0.001. (**C**) Scatter plot of *Dab1* mRNA expression and DAB1 protein expression in early developmental timepoints (GD21-PD21); Spearman Rho = 0.689, *p* < 0.001. (**D**) Scatter plot of *Dab1* mRNA expression and DAB1 protein expression in late developmental timepoints (PD35–100); Spearman Rho = −0.301, *p* = 0.021. Abbreviations: VEH, vehicle, MIA, maternal immune activation; M, male; F, female; GD, gestational day; PD, postnatal day.

**Figure 6 biomolecules-13-00489-f006:**
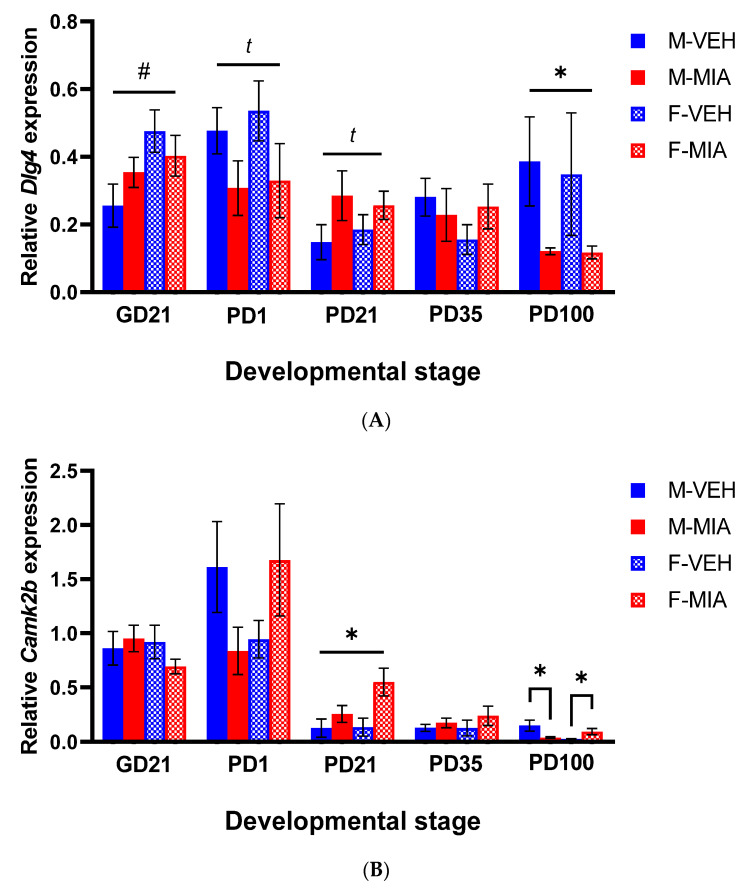
*Dlg4* (PSD-95) and *Camk2b* (CAMKII) gene expression across development. (**A**) Relative *Dlg4* (PSD-95) gene expression, calculated by normalising *Dlg4* expression to the geometric mean of three stable reference genes (*Gapdh, Ubc, B2m*). (**B**) Relative *Camk2b* (CAMKII) expression, calculated by normalising *Camk2b* expression to the geometric mean of three stable reference genes (*Gapdh, Ubc, B2m*). Bars represent mean ± SEM. Results of the GLMM (N = 5–7; n = 5–9) shown: effect of group, *t p* < 0.075; effect of group, * *p* < 0.05; effect of sex, # *p* < 0.05. Abbreviations: VEH, vehicle, PIC, poly(I:C); M, male; F, female; GD, gestational day; PD, postnatal day.

**Figure 7 biomolecules-13-00489-f007:**
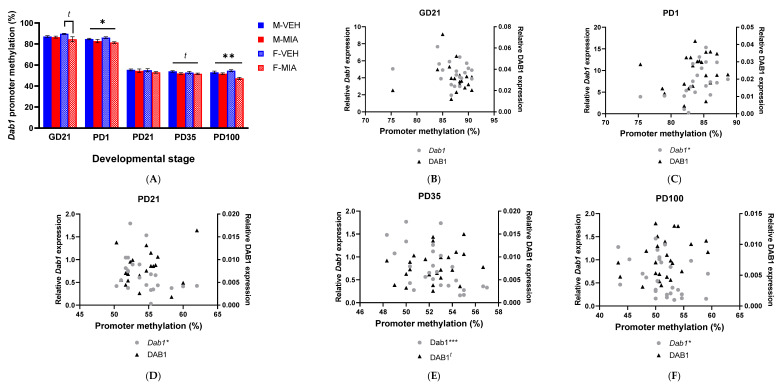
*Dab1* promoter methylation. (**A**) *Dab1* promoter methylation (%). Bars represent mean ± SEM. Results of the GLMM (N = 5–7; n = 5–7) shown: trending effect of group, *t p* < 0.075; effect of group, * *p* < 0.05, ** *p* < 0.01; abbreviations: VEH, vehicle, MIA, maternal immune activation; M, male; F, female; GD, gestational day; PD, postnatal day. (**B**–**F**) Correlation between *Dab1* promoter methylation (%) and *Dab1* mRNA expression (left Y-axis) and DAB1 protein expression (right Y-axis). (**B**) GD21 FC; (**C**) PD1 FC; (**D**) PD21 PFC; (**E**) PD35 PFC; (**F**) PD100 PFC. Results of Spearman rank correlation are indicated, * *p* < 0.05; *** *p* < 0.001, *^t^ p* < 0.075.

**Figure 8 biomolecules-13-00489-f008:**
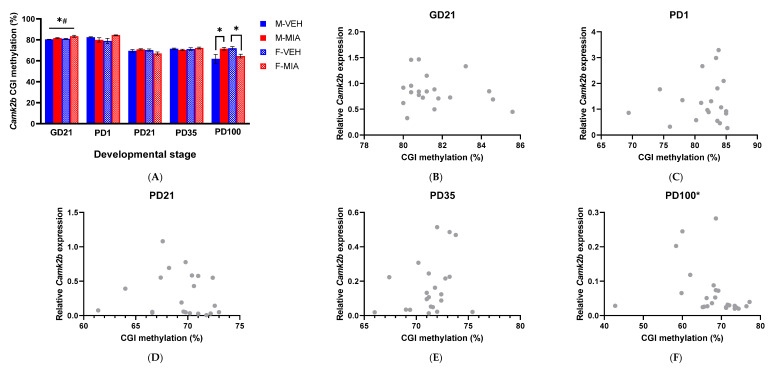
*Camk2b* CGI methylation. (**A**) *Camk2b* CGI methylation (%). Bars represent mean ± SEM. Results of the GLMM (N = 5–7; n = 5–7) shown: effect of group, * *p* < 0.05; effect of sex # *p* < 0.05; abbreviations: VEH, vehicle, MIA, maternal immune activation; M, male; F, female; GD, gestational day; PD, postnatal day. (**B**–**F**) Correlation between *Camk2b* CGI methylation (%) and *Camk2b* RNA expression (**B**) GD21 FC; (**C**) PD1 FC; (**D**) PD21 PFC; (**E**) PD35 PFC; (**F**) PD100 PFC. Results of Spearman rank correlation are indicated, * *p* < 0.05.

**Table 1 biomolecules-13-00489-t001:** Designed pyrosequencing assays. Pyrosequencing assays were designed using the Pyromark assay design software (v2.0, Qiagen, Manchester UK). The genic target sequence from the Rat Genome assembly Rn5.0 was entered into the assay design software, and the corresponding primer sequences for bisulphite-PCR and pyrosequencing were generated, alongside the sequence to analyse. The score of the assay reflects the probability the designed assay will work under standard conditions, with scores >70% considered good. Abbreviations: FP, forward primer (PCR); RP, reverse primer (PCR), * indicates the biotinylated PCR primer; SP, sequencing primer; Y/R indicates a CpG site within the sequence to analyse.

Gene Target	Primer	Sequence (5′-3′)	Sequence to Analyse	Score
*Dab1*	FP	TGAAATGTTTTTGTTGGTGTATGT	TATGTGYGGTTYGGGGTGTTTTTT TTGAAGGGAGGAGTTTTTTTTTTG GAGAGGATTTTYGATGAGTT TGGTTAAGGTT	71%
	RP *	AACCCCCAACCTTAACCAAACTC		
	SP	TGTAGTATTTAGATAGAGTGAATGA		
*Camk2b*	FP *	TTTTTGGGGGGTAAATTTAAGTG	CRAAACCRTCAACAACR ACRACTTTAAAAC CTATACRTAAATCTCCC	81%
	RP	ACCTAATAATCATCCCTATTTTCTCC		
	SP	ACCACAAAACAACTCA		

**Table 2 biomolecules-13-00489-t002:** OPT item preference pairwise comparisons across phases. The table shows pairwise comparisons between exploration times for each item in both the acquisition and retention phases. * = *p* < 0.05 after correction. *** = *p* < 0.001 after correction.

	Acquisition (Mean Difference, Y-X)				Retention (Mean Difference, Y-X)			
	Item 1 (Can/Bottle)	Item 2 (White Bottle)	Item 3 (Ceramic Pot)	Item 4 (Ramekin)	Item 1 (Can/Bottle)	Item 2 (White Bottle)	Item 3 (Ceramic Pot)	Item 4 (Ramekin)
**Item 1** **(Can/Bottle)**		−5.587, *p* < 0.001 ***	−2.350, *p* = 0.319	−10.045, *p* < 0.001 ***		−3.873, *p* < 0.001 ***	−2.965, *p* = 0.039 *	−14.544, *p* < 0.001 ***
**Item 2** **(White Bottle)**	5.587, *p* < 0.001 ***		3.237, *p* = 0.086	−4.457, *p* = 0.085	3.873, *p* < 0.001 ***		0.909, *p* = 1.000	−10.670, *p* < 0.001 ***
**Item 3** **(Ceramic Pot)**	2.350, *p* = 0.319	−3.237, *p* = 0.086		−7.696, *p* < 0.001 ***	2.965, *p* = 0.039 *	−0.909, *p* = 1.000		−11.579, *p* < 0.001 ***
**Item 4** **(Ramekin)**	10.045, *p* < 0.001 ***	4.458, *p* = 0.085	7.696, *p* < 0.001 ***		14.544, *p* < 0.001 ***	10.670, *p* < 0.001 ***	11.579, *p* < 0.001 ***	

## Data Availability

Data is available on request.

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
