# Peer review of "Maternal Immune Activation Induces Adolescent Cognitive Deficits Preceded by Developmental Perturbations in Cortical Reelin Signalling"

_biomolecules, 2023, doi:10.3390/biom13030489_

Round 1

Reviewer 1 Report

The study by Woods, Loruso et al. describes experiments in a PolyI:C induced MIA model of neurodevelopmental disorder.
The study has a very thorough theoretical introduction with a review of the current literature, even featuring Figure 1 illustrating Reelin signalling pathways. Having figures in the introduction is not common, however, it is very helpful in this case, as some readers may not be familiar with the reelin pathway and its ontogeny.
Concerning the methods, my expertise is mainly the behavioral tasks, so I cannot assess the gene and protein expression evaluation methods in sufficient depth, but overall the methodology and statistics seem sound and well described. It is very nice to see that not only the numbers of animals per group, but also the numbers of litters per group are indicated, which is crucial information in studies of MIA. The authors also mention negative results, e.g. in phospho-DAB non-detection or the more or less inconclusive data from the OPT. I consider reporting negative results a very positive sign, as those might be particularly helpful to other experimenters.
Overall, the article is well written in comprehensible English, and the results and their implications are well described.

I have only some minor recommendations for the manuscript:

Figure recommendations:

Figure 1 - add dendrite development to the schematic, as it seems an important outcome of reelin signalling.

Figure 4: the same color and pattern coding of groups should be used for both panels A and B for easier readability (make the bars of female groups chequered in both panels)

Having a schematic of the experiment timeline, illustrating the exact timing (polyI:C injection, brain sampling, behavioral testing) would help the reader to follow the course of the experiments.

Minor comments:

line 109-110 - there is an error report from reference manager, which should be removed and the references checked

line 159, in the the reference no. 15 the "robust cogitve deficit" was only present in females in the ASST if I read it correctly, this should be emphasized

line 173 - the subsection title begins with the word "subsection", presumably a leftover from editorial work on the manuscript, this should be removed.

Individually ventilated cages (IVC) were used for housing the experimental rats. Did they have any sort of enrichment, such as shelters, gnawing materials etc.? Different size of social groups in males (2-3) and females (5) should also be discussed at least briefly. In my experience, the size of the social group and housing conditions influence behavioral phenotypes to a great extent. IVC effectively isolates the animals from any outside stimulation (olfactory, acoustic), and it seems this may affect the phenotype manifestation of some neurodevelopmental models. Therefore, any detail of housing conditions is very important.

Recommendations for follow-up experiments and general comments:

A weak point of the present study is that cognition was not assessed at multiple points, its developmental course was not followed. As schizophrenia is characterized by a (relatively) latent period and usually becomes manifest at adolescence or early adulthood, one time point of testing is not enough. Also, using using multiple different behavioral paradigms would help interpretation of the results (e.g. was the relative lack of preference in the OPT a true failure of memory, or simply insufficient motivation to prefer the relocated objects?) and may provide better picture of the different dimensions of the phenotype. As it is, the paper clearly demonstrates a neurodevelopmental disorder, but the affinity towards schizophrenia is more or less an assumption. Maternal immune activation seems to be a common risk factor to both schizophrenia and autism spectrum disorders, with both disorders affecting Reelin signalling, cognition and reaction to novelty. I don't see this ambiguity as a flaw, however, trying to disentangle particular ASD-like and schizophrenia-like traits might be relevant for understanding the model and the underlying mechanisms.

For future studies, I would recommend testing at multiple timepoints to complement the developmental data for reelin pathway gene expression, optimally also using multiple tests. The hypothesised impacts of reelin dysregulation on synaptic plasticity and NMDA receptor functions should also be verified in a follow-up in this model.

It is also a pity that brain sampling was not done directly after the behavioral testing, to see what was happening in the brain at that particular age. It would even better to see direct correlations between behavior and reelin pathway dysregulation, which is particularly important as the phenotype might vary between individuals, as is the case with human schizophrenia - group-level statistics might blur the differences. Also, this would be a strong argument for causal relationship between the molecular and behavioral level changes.

Reviewer 2 Report

This study showed cognitive deficits in adolescent object-location memory in MIA-offspring and reductions in Reln expression prenatally and in the adult prefrontal cortex. This study also showed Reln, Dab1, Dlg4, Camk2b expression, and Dab1 promotor and Camk2b CGI methylation status in different developmental stages.

Because Reln has been shown to involved in the regulation of neuronal migration, dendritic growth and branching, synaptogenesis and synaptic plasticity. Reln also rescued MIA-induced cognitive deficits. Thus, the major limitation (weakness) of this study is that lacking the mechanistic evidence to connect (link) these observed altered gene expression profile with their cognitive function deficit. Which gene and in which developmental stage that matters the outcome of cognitive function. 

Reviewer 3 Report

In this study, the authors propose that maternal immune activation (MIA)-induced cognitive deficits in learning and memory and dysregulation of Reelin signaling contributes to pathophysiology of neurodevelopmental disorders like schizophrenia. Following their previous work, which validated a robust PFC-mediated cognitive deficit in adult rats exposed to MIA [reference 15], their aim was to establish the role of the Reelin pathway in contributing to cognitive deficit. In the present study, the authors used MIA rat model to demonstrated cognitive deficits in MIA-offspring and reductions in Reln expression prenatally and in the adult prefrontal cortex. They also assayed gene/protein expression and DNA methylation of downstream signaling components of Reelin that occurred subsequent to MIA-induced Reelin dysregulation and prior to cognitive deficits as proposed molecular mechanisms by which cognitive deficits could occur. The authors used an appropriate study design and standard methodology.

I have suggestions by following.

In some results, MIA causes sex-specific differences of gene expression changes and cognitive deficit in offspring. Data from their model seem to have sex-specific effects. More recently, it causes attention to sex differences in the brain development, gene expression, and epigenomic profile of schizophrenia. Thus, I suggest that the authors could explain in more detail about sex-specific effects in their findings and pathophysiology of schizophrenia in discussion part.

I suggest the authors should add limitation in the paper. For example,

A limitation is the lack of direct evidence for a causal relationship between MIA-induced Reln abnormality and alterations in offspring cognition. To address this, future studies should employ pharmacological approaches, such as to block Reln, to establish whether such deficits remain or employ rescue experiment to establish whether such deficits disappear.

Miner

In introduction: I suggest that the authors can briefly introduce polyinosinic:polycytidylic acid (poly(I:C).

Page 3 line 109-110. Fix this sentence (Error! Reference source not found.A).

Page 4 line 173. Delete “Subsection”

In figure 2, 3 legend, the authors should list VEH full name.

Round 2

Reviewer 2 Report

The author described the limitations in the discussion. Since the multiple pathways has been shown in MIA model, the links between the altered signaling in the brain and behavior deficit remains unclear.  
